# INFERRING REWARD FUNCTIONS
# FROM DEMONSTRATORS WITH UNKNOWN BIASES

## ABSTRACT

Our goal is to infer reward functions from demonstrations. In order to infer the
correct reward function, we must account for the systematic ways in which the
demonstrator is suboptimal. Prior work in inverse reinforcement learning can
account for specific, known biases, but cannot handle demonstrators with unknown
biases. In this work, we explore the idea of learning the demonstrator's *planning
algorithm* (including their unknown biases), along with their reward function.
What makes this challenging is that any demonstration could be explained either
by positing a term in the reward function, or by positing a particular systematic
bias. We explore what assumptions are sufficient for avoiding this impossibility
result: either access to tasks with known rewards which enable estimating the
planner separately, or that the demonstrator is sufficiently close to optimal that
this can serve as a regularizer. In our exploration with synthetic models of human
biases, we find that it is possible to adapt to different biases and perform better
than assuming a fixed model of the demonstrator, such as Boltzmann rationality.

## 1 INTRODUCTION

Our ultimate goal is to enable the design of agents that optimize for the *right* reward function. Unfor-
tunately, designing reward functions is challenging (Amodei et al., 2017) and can have unintended
side-effects (Hadfield-Menell et al., 2017). Inverse Reinforcement Learning (Russell, 1998; Ng et al.,
2000; Abbeel & Ng, 2004) aims to bypass the need for reward design by learning the reward from
observed demonstrations of good behavior.

Existing IRL algorithms typically make the assumption that the demonstrator is either optimal, or
Boltzmann rational, i.e. taking better actions with higher probability (Ziebart et al., 2008; Finn et al.,
2016). However, there is a rich literature showing that humans are *not* optimal, and are biased in
*systematic* ways. Consider a grad student who starts writing a paper a month in advance, expecting
it to take two weeks, but then misses the deadline. Should we infer that they prefer to lose sleep to
pursue a deadline that they then miss? Of course not – this is a classic case of the *planning fallacy*
(Buehler et al., 1994). It is not clear what exactly went wrong – perhaps the grad student did not have
a good dynamics model (not realizing that Amazon spot instances can be suddenly terminated), or
they failed to account for the uncertainty in their dynamics model (expecting that it wouldn't happen
to them), or they optimized too much for short term reward (watching Netflix), and so on – we do not
know what the bias is. Ideally, even with this unknown bias, we would like our system to infer that
the grad student failed to plan appropriately, and that they would prefer to meet the deadline.

This leads to a tradeoff between the expressivity of our model of the demonstrator, and the ability of
the algorithm to learn the true reward function with few samples. On one extreme, if we know exactly
what bias the demonstrator has, we can account for it to infer the true reward function quickly and
accurately, as in Evans et al. (2016); Evans & Goodman (2015); Zheng et al. (2014); Majumdar et al.
(2017) (myopia and hyperbolic time discounting, sparse noise, and risk-sensitivity respectively). Even
suboptimal trajectories or failures (Shiarlis et al., 2016) can be thought of as a biased demonstrator,
where the bias is the specific model of failure. Unfortunately, people are complex, and it is unlikely
that we will know exactly which bias a person is displaying, or even the space of possible biases that
people might have. If we are incorrect about the bias, we will have a mis-specified model, and the
reward function will get garbage values that explain the "noise" resulting from the mis-specification
(Steinhardt, 2017; Steinhardt & Evans, 2017). On the other extreme, when the demonstrator could be

any function mapping reward functions to policies, it is impossible to learn the true reward function even with infinite data, because there are always alternative explanations for the observed policy (Armstrong & Mindermann, 2017; Christiano, 2015). We could consider asking the human to rate each demonstration (Burchfiel et al., 2016), but the human could be biased in these ratings as well.

In this paper, we focus on learning rewards when the demonstrator's bias is *unknown*. We seek to avoid impossibility results by introducing a set of assumptions that are weaker than assuming a known bias, the core of which is that the demonstrator plans their actions similarly in similar tasks. This leads us to investigate the idea that one could actually learn, rather than assume, the demonstrator's *planning algorithm* across multiple tasks, including any biases they may have.

We analyze the feasibility of this idea in two settings, via two algorithms. In both algorithms, we can navigate the tradeoff between model expressivity and sample efficiency through the choice of architecture for the differentiable planner we learn, and the space of reward functions to consider. In the first setting, we assume that we have access to the true reward functions in some subset of tasks – that means we can learn the demonstrator's planning algorithm on those tasks, after which we can infer rewards from demonstrations in new tasks. In the more realistic setting where we have no access to any reward function, we assume that the demonstrator is "close" to optimal. This is a weaker version of the assumption of Boltzmann rationality that still allows us to learn particular systematic biases. We operationalize this assumption by initializing the differentiable planner to mimic the optimal agent, and then finetuning it to allow it to account for specific biases. We show that this initialization is essential for good performance, as we would expect from the impossibility result.

In summary, this paper makes three contributions:

1. Assumptions that are more realistic than the assumption of Boltzmann rationality that allow us to avoid known pitfalls of reward inference.

2. Algorithms that leverage these assumptions to infer reward functions from suboptimal demonstrations.

3. An experimental evaluation demonstrating there is hope in the idea of learning better rewards by learning biases too, despite the theoretical difficulty of the problem.

## 2 EXAMPLES OF BIASES

To put this work in context, we start with some examples of the kind of biases a general algorithm should be able to capture and account for. While we use these for illustrative purposes, the whole point of our work is that humans might have systematic suboptimalities completely different from these examples. We don't know all the possible biases a priori.

**Running Example.** We illustrate the effects of these biases on a simple 2D navigation task in figure 1. There are multiple salient locations, each of which has a desirability score (which can be negative, in which case the agent wants to avoid those locations). The agent can move in any of the four cardinal directions, or stay in its current position. Every movement action has a chance of failing and causing the agent to move in a direction orthogonal to the one it chose. Despite their simplicity, there are several ways in which human-like suboptimal behavior can manifest in these environments.

**Time inconsistency.** Would you prefer to get $100 in 30 days, or $110 in 31 days? Faced with this question, people typically choose the latter. However, thirty days later, when faced with the choice of getting $100 now, or $110 tomorrow, they sometimes choose to take the $100. This reversal of preferences over time would never happen with an optimal agent that maximizes expected sum of discounted rewards. Researchers model this phenomenon using *hyperbolic time discounting*, in which future rewards are discounted more aggressively than exponentially. This leads to a followup question – how do humans make long-term plans, given that their future self will have different preferences? Prior work has considered a spectrum from *naive* agents that assume their future self will have the same preferences as they do, to *sophisticated* agents that perfectly understand how their preferences will change over time and make plans that take such change into account (Frederick et al., 2002).

In figure 1, when going to a high reward, both the *naive* and *sophisticated* hyperbolic time discounters can be "tempted" by a proximate smaller reward. The naive agent fails to anticipate the temptation, and so once it gets near the smaller positive reward, it caves in to the temptation and stays there. The

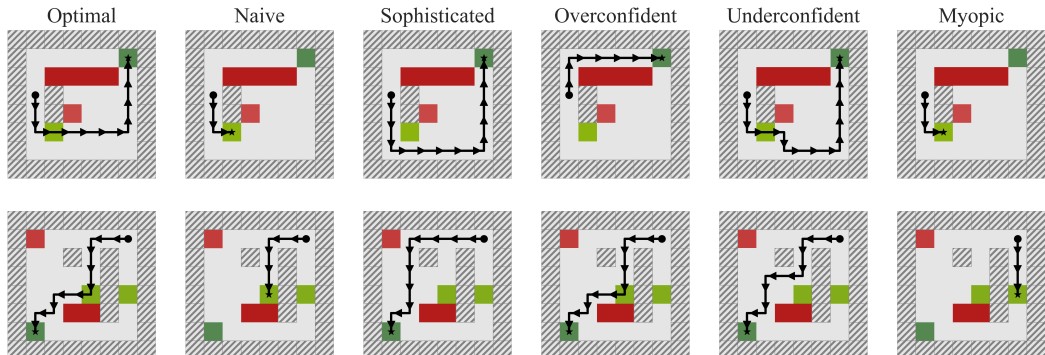

**Figure 1:** The plans of our synthetic agents on two navigation environments. Actual trajectories could differ due to randomness in the transitions. Green squares indicate positive reward while red squares indicate negative reward, with darker colors indicating higher magnitude of reward.

sophisticated agent explicitly plans to avoid the temptation – it does not collect the smaller reward and instead takes a longer, more dangerous path around the smaller reward to get to the large reward.

**Incorrect estimates of probabilities.** Humans are notoriously bad at judging probabilities. The *availability heuristic* (Tversky & Kahneman, 1973) refers to the human tendency to rate events as more likely if they are easier to recall. The *recency effect* is a similar effect where recent events are judged to be more probable. These biases are complicated and depend heavily on context and don't obviously transfer to our task. So, we use two simplified models – an *overconfident* agent, which expects that the most likely next state is more likely than it actually is, leading it to take risks, and an *underconfident* agent, which is overly cautious. In figure 1, the overconfident agent takes the shortest path to the reward, underestimating the risk of slipping into the large region of negative reward, while the underconfident agent plans a circuitous route around negative reward that it is unlikely to have actually encountered.

**Bounded computation.** Researchers have studied models of *bounded rationality*, where humans are assumed to be rational subject to the constraint that they have a bounded amount of computation. This can be thought of as an explanation that many other heuristics and biases are actually computational shortcuts that allow us to reach reasonably good decisions without too much cost (Kahneman, 2003). In our task, we model computation bounds as a small time horizon for planning, leading to *myopic* behavior. In figure 1, the myopic agent can only see close rewards, and goes directly to them, never even realizing the possibility of going to the highest reward.

## 3 PROBLEM: LEARNING REWARDS OF DEMONSTRATORS WITH UNKNOWN BIASES

**Notation.** A (finite-horizon) Markov Decision Process (MDP) (Puterman, 2014) is a tuple $\langle S, A, T, r, H \rangle$. $S$ is a set of states. $A$ is a set of actions. $T$ is a probability distribution over the next state, given the previous state and action. We write this as $T(s_{t+1}|s_t, a)$. $r$ is a reward function that maps states and actions to rewards $r : S \times A \to \mathbb{R}$. $H \in \mathbb{Z}_+$ is the finite planning horizon for the agent. Since we are interested in the setting where the reward function $r$ is unknown, we will factor MDPs into *world models* $w = \langle S, A, T, H \rangle$ and reward functions $r$.

Instead of having access to a reward function, we observe the behavior of an demonstrator, who performs the task well but may be suboptimal in systematic ways. In particular, we observe the demonstrator's *policy* $\pi : S \to A$.

**Estimating Biases and Rewards.** Given a ***world model*** $w$ and the ***demonstrator's policy*** $\pi_D$ which may exhibit an unknown bias, determine the ***reward*** $r^*$ that the demonstrator is optimizing.

We might hope to solve this problem without any additional assumptions. However, this problem is unsolvable – Armstrong & Mindermann (2017) prove an impossibility result showing that for any potential reward function $r'$, there is some planner $D'$ such that $D'(w, r') = D(w, r^*)$. The proof is simple – simply set $D'(w, r') = \pi$ for any $r'$, that is $D'$ always returns $\pi$ regardless of reward.

Inverse reinforcement learning assumes that the demonstrator is (approximately) optimal to get around this issue. It is common to assume Boltzmann rationality, where the probability of an action is proportional to the exponent of its expected value (Baker et al., 2006), i.e. $P(a|s) \propto e^{Q(s,a)}$, where $Q$ is the optimal Q function that satisfies the Bellman equation:

$$Q(s,a) = R(s,a) + \gamma \sum_{s'} \left[ T(s'|s,a) \max_{a'} Q(s',a') \right] \tag{1}$$

However, we know that humans are systematically suboptimal, and so we would like to relax this assumption and try other, more realistic assumptions. The pathological solutions in the impossibility result occur partly because the demonstrator can have arbitrary behavior on different environments. While we certainly want the demonstrator to adapt to different environments, the *algorithm* that the demonstrator uses to determine their policy should stay fixed across similar environments. This imposes structure on the demonstrator's planner that can eliminate some possibilities.

**Assumption 1:** The demonstrator plans in the same way for sufficiently similar environments.

To formalize this, we assume that there is a space of world models $\mathbb{W}$ with the same set of states $S$ and actions $A$, and a space of reward functions $R \subseteq S \times A \to \mathbb{R}$. The demonstrator can plan a stochastic policy given a world model and reward, that is, they are modeled as $D : (\mathbb{W} \times R) \to (S \to A \to [0,1])$. We call $D$ the planning algorithm used by the demonstrator, or planner for short. Of course, if $D$ can be any function with this type signature, it can still map any arbitrary $(w, r)$ pair to any arbitrary policy $D(w, r)$, but we will further ensure that $D$ is simple (through regularization). Given a list of world models $W = [w_1 \ldots w_n]$ and reward functions $R = [r_1 \ldots r_n]$, we define $D(W, R)$ to be the list of the demonstrator's policies $[D(w_1, r_1) \ldots D(w_n, r_n)]$.

Note that this is a strong assumption and it does limit the scope of our work: while it is reasonable to believe that people plan in the same way for variations of the same task, they likely have different biases for different tasks, because they may have domain-specific heuristics. The setting of multiple tasks has been studied before (Gleave & Habryka, 2018; Dimitrakakis & Rothkopf, 2011; Choi & Kim, 2012), though not for the purpose of inferring systematic biases.

This assumption leads to a slightly easier problem, of recovering rewards from multiple tasks:

**Estimating Biases and Rewards for Multiple Tasks.** Given a list of world models $W$ and the demonstrator's policies $\Pi_D = D(W, R)$ which may exhibit an unknown bias, determine the list of reward functions $R$ (one for each $w \in W$) that the demonstrator was optimizing.

Since the person uses the same planner across all tasks, an agent can have an easier time recovering rewards for each task by leveraging the common structure across the tasks. This is especially appealing for agents that would get to observe people for some period of time before trying to assist them. Unfortunately, Assumption 1 is not sufficient to solve this problem. Consider the case where the demonstrator is optimal. Given the assumptions so far, we could infer that the demonstrator is *minimizing* expected reward, i.e. that the reward they are optimizing is $-r^*$, since that perfectly predicts $\pi_D$. This is very bad, as we could infer a reward that incentivizes the *worst* possible behavior!

When humans take action, we typically assume that they are doing something that is reasonable for achieving their goals, even if it is not optimal:

**Assumption 2a:** The demonstrator is "close" to optimal.

We explore this assumption in section 4.3 to solve the problem of estimating biases and rewards for multiple tasks. We also explore an alternative approach, based on the fact that we have strong priors about what humans are trying to optimize for, which allow us to infer how good they are at achieving their goals. If we see some tasks where we know the demonstrator's reward function and policy, we can infer that the demonstrator is not minimizing expected reward.

**Assumption 2b:** We know what reward function the demonstrator is optimizing for *some* tasks.

**Estimating Biases and Rewards with Access to Tasks with Known Rewards.** Given a list of world models $W$, a list of the demonstrator's policies $\Pi_D = D(W, R)$, a list of world models $W_{\text{known}}$ with known rewards $R_{\text{known}}$ and a list of the demonstrator's policies $\Pi_{\text{known}} = D(W_{\text{known}}, R_{\text{known}})$, determine the reward functions $R$ that $D$ was optimizing.

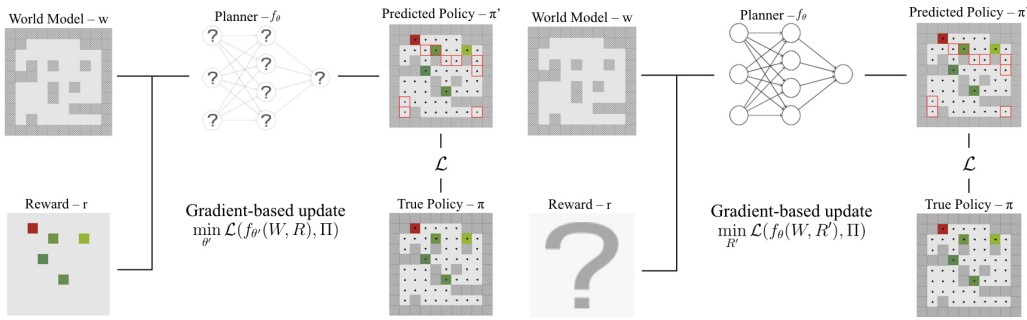

**(a)** Training the planner $f_\theta$. We hold the world model $w$, reward $r$, and policy $\pi$ fixed, and update the planner $f_\theta$ with gradient descent.

**(b)** Training the reward $R$. We hold the world model $w$, planner $f_\theta$, and policy $\pi$ fixed, and update the reward $r$ with gradient descent.

**Figure 2:** The architecture and operations on it that we use in our algorithms.

We will present algorithms for both settings in the next section. While our problem formulations above assume that we have access to full policies $\Pi_D$, none of our algorithms rely on this assumption – it is easy to modify them to work with trajectories instead.

## 4 ALGORITHMS TO ESTIMATE BIASES AND REWARDS

The key idea behind our algorithms is to learn a model of how the demonstrator plans, and invert the model's "understanding" using backpropagation to infer the reward from actions.

### 4.1 ARCHITECTURE

We model the demonstrator planning algorithm $D$ using a *differentiable planner* $f_\theta$, which is a neural net that can express planning algorithms whose parameters $\theta$ can be updated using gradient descent. $f$ has the same type as the demonstrator's planner $D$, namely $(\mathbb{W} \times \mathbb{R}) \to (S \to A \to [0, 1])$. Thus, the inputs to the differentiable planner $f$ are a world model $w \in \mathbb{W}$ and a reward function $r \in \mathbb{R}$; the output is a stochastic policy $\pi \in (S \to A \to [0, 1])$. We determine how well $\langle f, R \rangle$ matches the demonstrator's policy $\pi_D$ with the cross entropy loss $\mathcal{L}(f_\theta(W, R), \Pi_D) = \sum_i \mathcal{L}(f_\theta(w_i, r_i), \pi_{D,i})$.

While our algorithms can work with any differentiable planner, in this work we use a *value iteration network* (VIN) (Tamar et al., 2016). A VIN is a fully differentiable neural network that embeds an approximate value iteration algorithm inside a feed-forward classification network. For environments where transitions only depend on "nearby" states (as in navigation tasks), the Bellman update can be performed using an appropriate convolution, and the computation of values from Q-values can be done with a max-pooling layer. By leaving the filters for the convolutions unspecified, the VIN can automatically learn the transition probabilities. Of course, the VIN is merely one architecture for a differentiable planner; we could equally well use other planners (Srinivas et al., 2018; Pascanu et al., 2017; Guez et al., 2018). As research in this area advances, our work stands to benefit.

By choosing this particular architecture we are making another assumption:

**Assumption 3:** The inductive bias of $f_\theta$ is well-suited to the task of learning the planner $D$.

In our experiments, this assumption says that the demonstrator's planner is well-modeled by a Value Iteration Network. This is not a good assumption currently, but we expect that as more research is done on differentiable planners, they will become more sophisticated and will become better able to learn human planning algorithms.

**The components of our algorithms.** There are two main operations that we make use of on this architecture, which we illustrate in figure 2. First, given world models $W$, reward functions $R$ (either known or hypothesized), and the demonstrator's policies $\Pi_D$, we can train a corresponding planner using gradient descent (figure 2a):

$$\theta = \min_{\theta'} \mathcal{L}(f_{\theta'}(W, R), \Pi_D) \qquad \text{(TRAIN-PLANNER)}$$

Second, given world models $W$, demonstrator's policies $\Pi_D$, and some planner parameters $\theta$, we can infer the corresponding reward functions using gradient descent (figure 2b):

$$R = \min_{R'} \mathcal{L}(f_\theta(W, R'), \Pi_D) \qquad \text{(TRAIN-REWARD)}$$

We can also perform both of these at the same time by training the planner parameters and rewards jointly given world models $W$ and the demonstrator's policies $\Pi_D$:

$$R, \theta = \min_{R', \theta'} \mathcal{L}(f_{\theta'}(W, R'), \Pi_D) \qquad \text{(TRAIN-JOINTLY)}$$

### 4.2 LEARNING THE PLANNER FROM KNOWN REWARDS FIRST (ASSUMPTION 2B)

First, we tackle the simpler setting when we have access to a set of tasks with *known* rewards. We illustrate this in Algorithm 1. We first train the planner on the world models for which we have rewards, which lets us learn a model of how the demonstrator plans, including any systematic biases they may have. Then, we use the learned planner weights to infer the reward on the world models for which we don't know the reward.

---
**Algorithm 1** Estimating biases and rewards with access to tasks with known rewards.

---
1: **function** IRL-WITH-REWARDS($W$, $\Pi_D$, $W_{\text{known}}$, $R_{\text{known}}$, $\Pi_{\text{known}}$)
2:    $\theta \leftarrow$ TRAIN-PLANNER($W_{\text{known}}$, $R_{\text{known}}$, $\Pi_{\text{known}}$)
3:    **return** TRAIN-REWARD($W$, $\theta$, $\Pi_D$)

---

### 4.3 LEARNING THE PLANNER AND REWARDS SIMULTANEOUSLY (ASSUMPTION 2A)

The assumption that we have some tasks with known rewards is very strong, and may not hold in practice. How could we infer rewards without making this assumption? The obvious answer is to train $\theta$ and $R$ jointly as in Equation TRAIN-JOINTLY. However, since we have discarded Assumption 2b, we can once again learn the planner that minimizes expected reward, and infer that the reward is $-r^*$. We must instead use Assumption 2a, that the demonstrator is "close" to optimal.

To use this assumption in our algorithm, we simulate data from an optimal agent with randomly generated world models and rewards, and use this to train the planner to mimic an optimal agent. After initializing this way, we can then train the planner and reward jointly. This gives us Algorithm 2. We show in section 5.3 that the initialization, enacting assumption 2a, is crucial for good performance.

---
**Algorithm 2** Estimating biases and rewards for multiple tasks with no known rewards.

---
1: **function** IRL-WITHOUT-REWARDS($W$, $\Pi_D$)
2:    $W_{\text{sim}}, R_{\text{sim}} \leftarrow$ Generate random world models and rewards
3:    $\Pi_{\text{sim}} \leftarrow$ Run optimal agent on $\langle W_{\text{sim}}, R_{\text{sim}} \rangle$
4:    $\theta_{\text{init}} \leftarrow$ TRAIN-PLANNER($W_{\text{sim}}$, $R_{\text{sim}}$, $\Pi_{\text{sim}}$)
5:    $R_{\text{init}} \leftarrow$ TRAIN-REWARD($W$, $\theta$, $\Pi_D$)
6:    $\theta, R \leftarrow$ TRAIN-JOINTLY($W$, $\Pi_D$)         $\triangleright$ using $\theta_{\text{init}}$ and $R_{\text{init}}$ as initializations
7:    **return** $R$

---

## 5 EVALUATION

We evaluate our algorithms by simulating demonstrators with different biases, and testing whether the same method can correctly infer reward for all these demonstrators. Details on the experiment setup are provided in the supplementary material.

### 5.1 EVALUATING REWARD INFERENCE

**Hypothesis.** The key idea behind this work is that accounting for unknown systematic bias should outperform the assumption of a particular inaccurate bias, e.g. noisy rationality or the lack thereof.

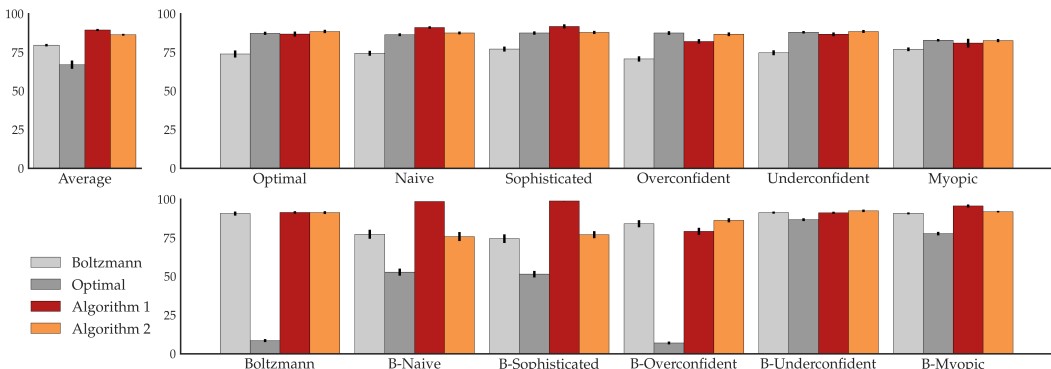

**Figure 3:** Reward obtained when planning with the inferred reward, as a percentage of the maximum possible reward, for different bias models and algorithms.

**Manipulated variables.** In order to test this, we manipulate whether we *learn* the demonstrator model or *assume* it. To avoid confounds introduced by changing the inference algorithm, we use the same algorithm for both. In the learning case, we train the planner on the ground truth demonstrator data; in the assume case, we train it on data generated from a) a Boltzmann-rational demonstrator; and b) an optimal demonstrator – these are the two models commonly assumed by IRL algorithms. Keeping the algorithm the same enables us to isolate the effect of adapting to an unknown model from the effect of having to use an approximate differentiable planner rather than a perfect one. We will quantify the second effect, i.e. the approximation error introduced by the VIN, in section 5.2

In the setting where we learn the bias, we further manipulate whether we have access to known rewards for some tasks or not – i.e. whether we use Algorithm 1 or Algorithm 2.

Finally, we manipulate the *actual bias* of the demonstrator. Following Evans et al. (2016), we implement the myopic, naive and sophisticated synthetic demonstrators as modifications of the value iteration algorithm. Similarly, we implement the overconfident and underconfident demonstrators by modifying the transition probability distributions used to plan in value iteration. We also include an optimal demonstrator, and stochastic (Boltzmann) versions of all demonstrators.

**Dependent measures.** We measure the reward obtained by planning optimally with the inferred reward function, as a percentage of the maximum possible reward that could be obtained.

**Findings.** Figure 3 shows our results comparing learning a demonstrator model with assuming an optimal or a Boltzmann demonstrator. The top left subfigure plots what happens on average, across all synthetic demonstrators we tested. The results provide support to our hypothesis: both learning methods (orange) outperform assuming a model (gray). Looking at the breakdown per demonstrator, we see that assuming optimal does not do well when the demonstrator has any noise (bottom graph). Similarly, assuming Boltzmann does not do well when the demonstrator is not noisy (top graph). The learning methods tend to perform on par with the best of two choices. In some cases, like the naive and sophisticated hyperbolic discounters, especially the noisy ones, the learning methods outperform both optimal and Boltzmann assumptions. The optimal assumption performs well in some of the non-noisy cases, because our demonstrator bias models are almost deterministic (they only break ties randomly). So, as long as the demonstrator eventually reaches the best reward location, the reward inference works well.

Interestingly, Algorithm 1 does not always outperform Algorithm 2, despite it having access to known rewards. We believe this has to do with the fact that Algorithm 2 exploits Assumption 2a (demonstrator close to optimal) and initializes from training on simulated optimal demonstrator data. Algorithm 1 does not rely on this assumption and therefore does not benefit from this initialization, whereas the assumption is correct for most of the models we test.

## 5.2 Tradeoff between being adaptive to bias vs. using exact planning

Our paper is not about a practical solution to be used right now, but rather an investigation of the viability of an idea. The core reason behind this is that to be adaptive to different kinds of biases we

**Figure 4:** Percent reward obtained for different bias models using variations of Algorithm 2, which does not get access to any known rewards. These algorithms can vary along two dimensions – whether they are initialized with the assumption that the demonstrator is rational, and whether they train the planner and reward jointly or with coordinate ascent. The original version of Algorithm 2 does initialize, and trains jointly.

might see, we learn a model of the demonstrator's planning algorithm via a differentiable planner. Unfortunately, this causes our planning to be approximate – whatever benefit we get from the adapting to biases, we lose because of the approximation. But these planners will become more practical, they can make this idea practical as well.

To quantify this loss, we replace the VIN with a differentiable exact model of the demonstrator, and infer the reward by backpropagating through the exact model. Since value iteration is not differentiable, we implement soft value iteration, where max operations are replaced with logsumexp operations, and measure percent reward obtained when inferring rewards for an optimal demonstrator.

**Results.** With an exact model of the demonstrator, we get $(98.1\pm0.1)\%$ of the maximum reward when performing optimal planning on the inferred rewards, while we get $(86.9 \pm 1.6)\%$ with Algorithm 1 and $(86.2 \pm 1.6)\%$ with Algorithm 2. Again, better planners would improve both algorithms.

### 5.3    HOW IMPORTANT ARE THE VARIOUS PARTS OF THE ALGORITHM?

Algorithm 2 was predicated on Assumption 2a, that the demonstrator's planner was "close" to rational, which motivated the initialization step where the planner is trained to mimic an optimal agent. We test how important this is by modifying Algorithm 2 to infer rewards without an initialization (removing lines 2-5). We include versions of the algorithm where we perform coordinate ascent by alternating planner training and reward training instead of training the planner and reward jointly.

**Results.** Figure 4 shows the results for a subset of demonstrators (full results are in the supplementary material). We can see that the initialization is indeed crucial for good performance, as expected. It also turns out that the joint training outperforms coordinate ascent.

## 6    DISCUSSION

**Summary.** In this work, we considered the very general problem of inverse reinforcement learning when the demonstrator has an unknown bias. In this problem we face a severe tradeoff between the ability to adapt to unknown biases, and the ability to infer any reward function. We introduced assumptions that are weaker than the typical assumption of noisy rationality, and tested algorithms that leverage these assumptions to perform better on average across different kinds of biases than assuming a specific bias known a priori.

**Limitations and future work.** We see this paper as taking a step towards exploring the idea of learning planners from demonstrations to learn reward functions. This is not yet a practical idea, due to the practical limitations of differentiable planners. Thus, to analyze how feasible this is with different possible biases and with ground truth reward functions, we resorted to synthetic demonstrator data as opposed to human data.

Further, our assumption that the demonstrator has the same bias across many tasks is key to our work, but is very strong. We could extend this work by using meta-learning to learn a prior over planners. We will also need to infer the demonstrator's beliefs as in Baker & Tenenbaum (2014), for which we could use TOMNets (Rabinowitz et al., 2018). We are excited to look into this, and into what additional inductive bias we could leverage, in our future work.

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

## A  ADDITIONAL EXPERIMENTAL DATA

In section 5, we presented data on the results of running various algorithms against a set of demonstrators, reporting the reward obtained according to the true reward function when using the inferred reward with an optimal planner, as a percentage of the maximum possible true reward. Table 1 shows the percentage reward obtained for all combinations of algorithms and demonstrators. We also measure the accuracy of the planner and reward at predicting the demonstrator's actions in new gridworlds where the rewards are the same but the wall locations have changed. These results are presented in Table 2. Note that there are often multiple optimal actions at a given state, which makes it challenging to get high accuracy.

**Table 1:** Percent reward obtained when the algorithm (column) is used to infer the bias of the demonstrator (row). The optimal and Boltzmann algorithms assume a fixed model of the demonstrator and train the VIN to mimic the model before performing reward inference (and were used in figure 3). We also include the four flavors of Algorithm 2 that were plotted in figure 4. The VI algorithm uses a differentiable implementation of soft value iteration as the planner instead of a VIN (used in section 5.2). The demonstrators are the optimal agent, the biased agents of figure 1, and versions of each of these agents with Boltzmann noise.

| Agent | Optimal | Boltzmann | Algorithm 1 | Coord w/ init | Joint w/ init | Coord w/o init | Joint w/o init | VI |
|---|---|---|---|---|---|---|---|---|
| Average | $67.0 \pm 2.7$ | $79.7 \pm 0.8$ | $89.5 \pm 0.7$ | $85.0 \pm 0.5$ | $86.4 \pm 0.6$ | $-3.9 \pm 0.7$ | $2.6 \pm 1.0$ | $71.9 \pm 3.0$ |
| Optimal | $87.3 \pm 1.0$ | $73.9 \pm 2.3$ | $86.9 \pm 1.6$ | $86.2 \pm 1.6$ | $88.5 \pm 1.1$ | $-4.2 \pm 1.2$ | $2.6 \pm 3.7$ | $98.1 \pm 0.1$ |
| Naive | $86.4 \pm 0.9$ | $74.4 \pm 1.6$ | $91.1 \pm 0.8$ | $84.6 \pm 1.2$ | $87.5 \pm 0.9$ | $-3.2 \pm 1.3$ | $2.6 \pm 3.7$ | $96.1 \pm 0.1$ |
| Sophisticated | $87.5 \pm 1.1$ | $77.1 \pm 1.6$ | $91.8 \pm 1.3$ | $83.6 \pm 1.3$ | $87.9 \pm 1.0$ | $-3.6 \pm 1.4$ | $2.6 \pm 3.7$ | $96.7 \pm 0.1$ |
| Myopic | $82.8 \pm 0.8$ | $77.0 \pm 1.2$ | $81.0 \pm 2.8$ | $80.6 \pm 0.8$ | $82.6 \pm 1.0$ | $-5.5 \pm 2.8$ | $2.6 \pm 3.7$ | $87.5 \pm 0.2$ |
| Overconfident | $87.5 \pm 1.2$ | $70.7 \pm 1.7$ | $82.1 \pm 1.4$ | $83.9 \pm 1.5$ | $86.7 \pm 1.2$ | $-2.7 \pm 1.1$ | $2.6 \pm 3.7$ | $97.5 \pm 0.1$ |
| Underconfident | $88.0 \pm 0.8$ | $74.7 \pm 1.6$ | $86.7 \pm 1.2$ | $86.1 \pm 1.5$ | $88.5 \pm 1.0$ | $-2.4 \pm 1.4$ | $2.6 \pm 3.7$ | $98.9 \pm 0.2$ |
| Boltzmann | $8.5 \pm 1.0$ | $90.7 \pm 1.3$ | $91.4 \pm 0.8$ | $88.4 \pm 1.6$ | $91.3 \pm 0.9$ | $-3.0 \pm 1.9$ | $2.6 \pm 3.7$ | $8.7 \pm 0.1$ |
| B-Naive | $52.8 \pm 2.3$ | $77.3 \pm 2.9$ | $98.5 \pm 0.1$ | $82.5 \pm 2.4$ | $75.8 \pm 2.9$ | $-8.3 \pm 4.5$ | $2.6 \pm 3.7$ | $47.7 \pm 0.2$ |
| B-Sophisticated | $51.5 \pm 2.1$ | $74.5 \pm 2.8$ | $98.8 \pm 0.2$ | $80.1 \pm 1.5$ | $77.0 \pm 2.3$ | $-8.7 \pm 3.9$ | $2.6 \pm 3.7$ | $48.0 \pm 0.2$ |
| B-Myopic | $77.7 \pm 1.1$ | $90.8 \pm 0.6$ | $95.6 \pm 1.0$ | $91.5 \pm 0.6$ | $91.9 \pm 0.5$ | $-2.4 \pm 2.1$ | $2.6 \pm 3.7$ | $83.4 \pm 0.1$ |
| B-Overconfident | $7.0 \pm 0.9$ | $84.1 \pm 2.3$ | $79.2 \pm 2.3$ | $81.4 \pm 2.8$ | $86.3 \pm 1.3$ | $-0.8 \pm 1.6$ | $2.6 \pm 3.7$ | $8.7 \pm 0.1$ |
| B-Underconfident | $86.7 \pm 0.9$ | $91.3 \pm 0.7$ | $91.2 \pm 0.7$ | $90.7 \pm 1.0$ | $92.4 \pm 0.8$ | $-1.8 \pm 1.2$ | $2.6 \pm 3.7$ | $92.1 \pm 0.1$ |

**Table 2:** Accuracy when predicting the demonstrator's actions (row) on new gridworlds using the planner and reward inferred by the algorithm (column). Algorithms and demonstrators are the same as in Table 1.

| Agent | Optimal | Boltzmann | Algorithm 1 | Coord w/ init | Joint w/ init | Coord w/o init | Joint w/o init | VI |
|---|---|---|---|---|---|---|---|---|
| Optimal | $61.3 \pm 0.4$ | $59.8 \pm 0.4$ | $62.0 \pm 0.3$ | $62.8 \pm 0.2$ | $63.6 \pm 0.3$ | $63.0 \pm 0.2$ | $72.4 \pm 0.1$ | $25.7 \pm 0.1$ |
| Naive | $60.1 \pm 0.3$ | $59.4 \pm 0.3$ | $58.6 \pm 0.3$ | $61.3 \pm 0.3$ | $61.8 \pm 0.3$ | $61.0 \pm 0.3$ | $71.1 \pm 0.1$ | $24.9 \pm 0.1$ |
| Sophisticated | $60.5 \pm 0.4$ | $59.2 \pm 0.4$ | $59.3 \pm 0.3$ | $61.0 \pm 0.3$ | $62.0 \pm 0.4$ | $61.2 \pm 0.3$ | $71.2 \pm 0.1$ | $24.9 \pm 0.1$ |
| Myopic | $54.1 \pm 0.4$ | $53.5 \pm 0.5$ | $54.9 \pm 0.5$ | $55.6 \pm 0.2$ | $56.1 \pm 0.3$ | $56.0 \pm 0.1$ | $62.8 \pm 0.1$ | $20.4 \pm 0.1$ |
| Overconfident | $61.6 \pm 0.4$ | $60.1 \pm 0.4$ | $61.8 \pm 0.4$ | $63.3 \pm 0.3$ | $63.7 \pm 0.3$ | $63.1 \pm 0.2$ | $72.8 \pm 0.1$ | $25.9 \pm 0.1$ |
| Underconfident | $60.9 \pm 0.4$ | $59.5 \pm 0.4$ | $61.4 \pm 0.3$ | $62.4 \pm 0.3$ | $62.9 \pm 0.3$ | $62.5 \pm 0.3$ | $72.0 \pm 0.1$ | $25.5 \pm 0.1$ |
| Boltzmann | $56.7 \pm 1.1$ | $60.5 \pm 0.4$ | $60.9 \pm 0.3$ | $60.3 \pm 0.2$ | $60.8 \pm 0.3$ | $62.3 \pm 0.3$ | $67.1 \pm 0.5$ | $24.2 \pm 0.1$ |
| B-Naive | $56.6 \pm 0.8$ | $59.8 \pm 0.8$ | $60.4 \pm 0.1$ | $60.3 \pm 0.2$ | $60.5 \pm 0.7$ | $59.9 \pm 0.3$ | $68.5 \pm 0.3$ | $23.7 \pm 0.1$ |
| B-Sophisticated | $57.6 \pm 0.7$ | $60.2 \pm 0.7$ | $60.5 \pm 0.2$ | $60.5 \pm 0.2$ | $61.2 \pm 0.3$ | $60.1 \pm 0.3$ | $68.5 \pm 0.3$ | $23.7 \pm 0.1$ |
| B-Myopic | $56.3 \pm 0.2$ | $56.9 \pm 0.4$ | $55.9 \pm 0.2$ | $56.5 \pm 0.2$ | $57.0 \pm 0.2$ | $56.3 \pm 0.1$ | $62.4 \pm 0.1$ | $20.3 \pm 0.0$ |
| B-Overconfident | $56.9 \pm 1.1$ | $60.7 \pm 0.4$ | $61.3 \pm 0.3$ | $60.9 \pm 0.2$ | $61.6 \pm 0.3$ | $62.7 \pm 0.2$ | $68.0 \pm 0.5$ | $24.2 \pm 0.1$ |
| B-Underconfident | $62.4 \pm 0.3$ | $63.1 \pm 0.4$ | $63.4 \pm 0.2$ | $63.0 \pm 0.1$ | $63.6 \pm 0.1$ | $63.5 \pm 0.2$ | $72.2 \pm 0.1$ | $25.4 \pm 0.1$ |

## B  EXPERIMENT DETAILS

All results are averaged over 10 runs with different seeds, on randomly generated 14x14 gridworlds that have 7 squares with non-zero rewards. We ensure that all such squares can be reached from the start state, and that at least half of the positions in grid are not walls.

We use a Value Iteration Network with 10 iterations as our differentiable planner, and set the space of rewards to be $S \to \mathbb{R}$; that is, any state can be mapped to any reward, but the reward is assumed not to depend on the action. We added an extra convolutional layer to the initial part of the VIN (which learns the proxy reward) as initial experiments showed that this could better learn an optimal planner for our gridworlds. We apply L2 regularization to the VIN with scale 0.0001, and do not regularize the reward.

For all experiments, we kept the number of demonstrations fixed to 8000. For Algorithm 1, this was split into 7000 policies with rewards that were used to train the planner, and 1000 on which rewards had to be inferred. Note that this does *not* include any simulated data – for example, Algorithm 2 would get 8000 biased policies, and would *also* simulate a further 7000 policies from an optimal agent in order to initialize the planner and reward.

