# OpenReview forum: "Inferring Reward Functions from Demonstrators with Unknown Biases"
_ICLR.cc/2019/Conference_

### Official Review · AnonReviewer1 · 2018-11-03
**Excellent motivation of work but lacks technical merit; results not convincing**

**Rating:** 5
**Confidence:** 4

**Review:**

This paper has proposed algorithms for inferring reward functions from demonstrations with unknown biases. To achieve this, the authors have proposed to learn planners from demonstrations in multiple tasks via value iteration networks to learn the reward functions.

This paper has provided an excellent motivation of their work in Sections 1 & 2 with references being made to human behaviors and heuristics, though the authors can choose a more realistic running example that is less extreme than making orthogonal decisions/actions. The paper is well-written, up till Section 4.

On the flip side, there does not seem to be any significant technical challenges, perhaps due to some of the assumptions that they have made. Like the authors have mentioned, I do find assumption 3 to be overly strong and restrictive, as empirically demonstrated in Section 5.2. Arguably, is it really weaker than that of noisy rationality? At this moment, it is difficult to overlook this, even though the authors have argued that it may not be as restrictive in the future when more sophisticated differentiable planners are developed.

The experimental results are not as convincing as I would have liked. In particular, Algorithm 2 (learning a demonstrator's model) does not seem to outperform that of assuming an optimal demonstrator for the noiseless case and a Boltzmann demonstrator for the noisy case (Fig. 3). This was also highlighted by the authors as well: "The learning methods tend to perform on par with the best of two choices." It begs the question whether  accounting for unknown systematic bias can indeed outperform the assumption of a particular inaccurate bias when we know a priori whether the demonstrations are noisy or not.



Other detailed comments are provided below:

I would have preferred that the authors present their technical formulations in Section 4 using the demonstrator's trajectories instead of policies.

The authors say that "In some cases, like they naive and sophisticated hyperbolic discounters, especially the noisy ones, the learning methods outperform both optimal and Boltzmann assumptions." But, Fig. 3 shows that Algorithm 2 does not perform better than either that of the optimal or Boltzmann demonstrator.

In Section 5.2, the authors have empirically demonstrated the poor approximate planning performance of VIN, as compared to an exact model the demonstrator. What then would its implications be on the adaptivity of Algorithms 1 and 2 to biases?

The following references on IRL with noisy demonstration trajectories would be relevant:

Benjamin Burchfiel, Carlo Tomasi, and Ronald Parr. Distance Minimization for Reward Learning from Scored Trajectories. In Proc. AAAI, 2016.

J. Zheng, S. Liu, and L. M. Ni. Robust Bayesian inverse reinforcement learning with sparse behavior noise. In Proc. AAAI, 2014.



Minor issues:
On page 4, the expression D : W × R -> S -> A -> [0, 1] can be more easily understood with the use of parentheses.

For Assumption 2b, you can italicize "some".

In the first paragraph of section 4.1, what are you summing over?

Line 3 of Algorithm 1: PI_W?

Page 7: For the learning the bias setting?

Page 7: figure 3 shows?

Page 7: they naive?

Page 7: so as long as?

Page 8: adaption?

Page 8: predicated?

Page 8: figure 4 shows?

---

### Official Review · AnonReviewer3 · 2018-11-05
**Paper studies a relevant and interesting problem but needs extended empirical evaluation**

**Rating:** 5
**Confidence:** 3

**Review:**

This paper addresses the interesting and challenging problem of learning the reward function from demonstrators which have unknown biases. As this is in general impossible, the authors consider two special cases in which either the reward function is observed on a subset of tasks or in which the observations are assumed to be close to optimal. They propose algorithms for both cases and evaluate these in basic experiments.

The studied problem is relevant as many/most demonstrators have unknown biases and we still need methods to effectively learn from those.

As far as I am aware of the related literature, the problem has not been studied in that explicit form although there is related work which targets the problem of learning from suboptimal demonstrators or demonstrators that can fail, e.g. [1] (I suggest to discuss this and other relevant papers in a related work section).

The main shortcomings of the paper are a lack of clarity at certain points and a limited experimental validation:
* For instance, the formalization of „Assumption 1“ is unclear. In which sense does this cover similarity in planing? As far as I understand, the function D could still map any combination of world model and reward function to any arbitrary policy. What does it mean that the planning algorithm D is „fixed and independent“?
* A crucial point requiring more investigation in my opinion is Assumption 3 (well-suited inductive bias). Empirically the chosen experimental setup yields expected results. However, to better understand the problem of learning with unknown biases it would be important to see how results change if the model for the planner changes. A small step in that direction would have been to provide results for value iteration networks with different number of iterations and number neurons, etc.
* If you use the differentiable planner instead of the VIN, how many iterations do you unroll?
* Is there any evidence that the proposed approach can work effectively in larger scale domains with more difficult biases? Also in the case in which the biases are inconsistent among demonstrations?

Further suggestions:
* Test how algorithm 1 performs if first initialized on simulated optimal demonstrations.
* Improve notation for the planning algorithm D by using brackets.

[1] Shiarlis, K., Messias, J., & Whiteson, S. (2016, May). Inverse reinforcement learning from failure. In Proceedings of the 2016 International Conference on Autonomous Agents & Multiagent Systems (pp. 1060-1068). International Foundation for Autonomous Agents and Multiagent Systems.

---

### Official Review · AnonReviewer2 · 2018-11-08
**Interesting topic and approach, needs work and careful evaluation**

**Rating:** 5
**Confidence:** 4

**Review:**

Not all examples in the introduction are necessarily biases but can be modeled with reward functions, where reward is given to specific states other than finishing work by the deadline. It would be helpful for the reader to get examples that  correspond to the investigated biases.

It would be good if the authors could at least mention that “Boltzmann rational” is a specific model of “systematic” bias for which much experimental support eith humans and animals exists.

The authors are strongly encouraged to review the literature on IRL, which includes other examples of modeling explicitly suboptimal agents, e.g.:
- Rothkopf, C. A., & Dimitrakakis, C. (2011). Preference elicitation and inverse reinforcement learning. ECML.
Similarly, the idea to learn an agent’s reward functions across multiple tasks has also appeared in the literature before, e.g.:
- Dimitrakakis, C., & Rothkopf, C. A. (2011). Bayesian multitask inverse reinforcement learning. EWRL.
- Choi, J., & Kim, K. E. (2012). Nonparametric Bayesian inverse reinforcement learning for multiple reward functions. NIPS

The authors state:
“The key idea behind our algorithms is to learn a model of how the demonstrator plans, and invert the model’s "understanding" using backpropagation to infer the reward from actions.”
It would be also important in this case to relate this to prior work, as several authors have proposed a very similar idea, in which a particular parameterization of the agent’s planning given the rewards and the transition function are learned, including Ziebart et al. and Dimitrakakis et al. This is also related to
- Neu, G., & Szepesvári, C. (2007). Apprenticeship learning using inverse reinforcement learning and gradient methods. UAI.

It would be great if the authors could also discuss how assumption 3 is a necessary for accurately inferring reward functions and biases and how deviations from this assumption interfere with the goal of this inference. This seems to be a central and important point for the viability of the approach the authors take here.

Currently, the evaluation of the proposed method is in terms of the loss incurred by a planner between the inferred reward function and the true reward function, figure 3. It would be important for the evaluation of the current manuscript to know what the inferred biases are. That using a wrong model of how actions are generated given values, e.g. myopic vs. Boltzmann-rational, results in wrong inferences, should not be too surprising. Therefore, the main question is: does the proposed algorithm recover the actual biases?

Minor points:
“like they naive and sophisticated hyperbolic discounters”

---

### Meta-Review · Area_Chair1 · 2018-12-13
**Interesting idea but paper needs more work**

**Confidence:** 4
**Recommendation:** Reject

**Metareview:**

The authors study an inverse reinforcement learning problem where the goal is to infer an underlying reward function from demonstration with bias.  To achieve this, the authors learn the planners and the reward functions from demonstrations. As this is in general impossible, the authors consider two special cases in which either the reward function is observed on a subset of tasks or in which the observations are assumed to be close to optimal. They propose algorithms for both cases and evaluate these in basic experiments. The problem considered is important and challenging. One issue is that in order to make progress the authors need to make strong and restrictive assumptions (e.g., assumption 3, the well-suited inductive bias). It is not clear if the assumptions made are reasonable. Experimentally, it would be important to see how results change if the model for the planner changes and to evaluate what the inferred biases would be. Overall, there is consensus among the reviewers that the paper is interesting but not ready for publication.